# The Effectiveness of Heat-Killed *Pediococcus acidilactici* K15 in Preventing Respiratory Tract Infections in Preterm Infants: A Pilot Double-Blind, Randomized, Placebo-Controlled Study

**DOI:** 10.3390/nu16213635

**Published:** 2024-10-25

**Authors:** Kenichi Takeshita, Haruka Hishiki, Haruka Takei, Naho Ikari, Saori Tanaka, Yuta Iijima, Hitoshi Ogata, Kensuke Fujishiro, Takahiro Tominaga, Yuki Konno, Yukiko Iwase, Taiji Nakano, Mamiko Endo, Naruhiko Ishiwada, Yoshiteru Osone, Tomohiro Kawaguchi, Taro Horiba, Ryo Takemura, Hiromichi Hamada, Naoki Shimojo

**Affiliations:** 1Department of Pediatrics, Graduate School of Medicine, Chiba University, 1-8-1 Inohana, Chuo-ku, Chiba-shi 260-8670, Chiba, Japan; 2Research and Development Division, Kikkoman Corporation, 338 Noda, Noda-shi 278-0037, Chiba, Japan; 3Perinatal Medical Center, Chiba University Hospital, 1-8-1 Inohana, Chuo-ku, Chiba-shi 260-8677, Chiba, Japan; 4Department of Infectious Diseases, Medical Mycology Research Center, Chiba University, 1-8-1 Inohana, Chuo-ku, Chiba-shi 260-8673, Chiba, Japan; 5Clinical and Translational Research Center, Keio University Hospital, 35 Shinanomachi, Shinjuku-ku 160-8582, Tokyo, Japan; 6Center for Preventive Medical Sciences, Chiba University, 1-8-1 Inohana, Chuo-ku, Chiba-shi 260-8670, Chiba, Japan; shimojo@faculty.chiba-u.jp

**Keywords:** heat-killed *Pediococcus acidilactici* K15, preterm infant, respiratory tract infections, lactic acid bacteria, salivary s-IgA, fecal microbiota

## Abstract

Background: Preterm infants discharged from the neonatal intensive care unit (NICU) have a risk of severe viral respiratory tract infections (RTIs). Researchers have recently reported the potential use of postbiotics to decrease RTIs in young children. However, the safety and efficacy of postbiotics for preventing RTIs in preterm infants is not yet established. Methods: We conducted a pilot double-blind, randomized, placebo-controlled study of the heat-killed lactic acid bacterium *Pediococcus acidilactici* K15 in 41 preterm infants born at <36 weeks of gestation and discharged from the NICU at Chiba University Hospital. Results: Following once-daily K15 or placebo treatment for one year, no significant differences were found in the mean number of febrile days (4.5 [1.5–7.4] days vs. 6.6 [2.6–10.5] days). The subgroup analysis showed that the effect of treatment on the number of febrile days was more prominent in the K15 group than in the placebo group, among children with older siblings. The 16S rRNA gene sequencing of fecal samples illustrated that the genus *Faecalimonas* was enriched in the K15 group, potentially promoting butyrate production by butyrate-producing bacteria. No adverse events were found to be associated with K15 intake. Conclusion: There were no clear data to show the effectiveness of K15 in preventing fever and RTIs in preterm babies during infancy. A larger clinical trial is warranted.

## 1. Introduction

In recent years, the improvement in the quality of neonatal care has been remarkable, and Japan has the lowest rate of neonatal death globally. However, the preterm birth rate has increased. Preterm infants have an immature immune system and receive lower maternal antibody levels than full-term infants [1,2]. Preterm infants have a risk of severe viral respiratory tract infections (RTIs), such as respiratory syncytial virus (RSV) infection, after discharge from the neonatal intensive care unit (NICU) [3]. Preterm infants also have an immature intestinal microbiota, which can result in immaturity-related difficulties regarding infections [4,5].

Probiotics confer multiple beneficial effects on the human intestinal microbiota. Some of them, for example, Lactic Acid Bacteria (LAB), activate the immune system and help prevent bacterial and viral infections [6,7,8,9]. Meanwhile, postbiotics, defined by the International Scientific Association for Probiotics and Prebiotics (ISAPP) as “a preparation of inanimate microorganisms and/or their components that confers a health benefit on the host” also have recently been recognized as playing a role in achieving health benefits [10]. *Pediococcus acidilactici* is one of the LABs found in fermented foods and the human digestive tract, and it can potentially modulate the immune system by inducing the production of type I interferons (IFNs) in common carp [11]. In particular, *P. acidilactici* K15 is a strain isolated from a fermented rice bran bred to achieve high immune activation ability, and even after heat sterilization, it still shows its potential as a postbiotic. In our previous study, heat-killed K15 induced the production of IFN-β by human blood-derived dendritic cell antigen 1 (BDCA1)-positive dendritic cells (DCs) and high levels of IgA production by human B cells stimulated by BDCA1-positive DCs in a cell culture model [12,13].

Moreover, in our previous clinical study, a randomized controlled study of preschool children revealed that heat-killed K15 (5 × 10^10^ bacteria) given every day for sixteen weeks supported anti-infectious immune systems in children who consumed less fermented foods and promoted IgA secretion in saliva [14]. We hypothesized that the administration of heat-killed K15 could activate the immune system of preterm infants and help prevent ARIs. Additionally, we hypothesized that this may be safer than live probiotics, which carry a risk of translocation in premature infants. However, there is currently no evidence regarding the safety and effectiveness of heat-killed K15 for preventing RTIs in infants, particularly in preterm infants. Therefore, we conducted a pilot double-blind, randomized, placebo-controlled study of heat-killed *P. acidilactici* K15 in preterm infants after their NICU discharge.

## 2. Materials and Methods

### 2.1. Preparation of Clinical Test Foods

Heat-killed *P. acidilactici* K15 powder was prepared by the Kikkoman Corporation (Chiba, Japan). K15 was cultured in a medium containing soy peptide, yeast peptone, glucose, and sodium acetate for 24 h. Then, K15 was heat-killed at 90 °C, washed with saline using ultrafiltration columns, and spray-dried with dextrin. The test food for the K15 group was prepared by blending 9.1 mg of K15 powder (5 × 10^10^ bacteria) with 1 g of dextrin powder. For the placebo group, 1 g of dextrin powder was packed alone.

### 2.2. Subjects and Clinical Study Design (Figure 1)

A randomized, double-blind, placebo-controlled trial was conducted at Chiba University Hospital (Chiba, Japan). Preterm infants who were born at <36 weeks of gestation and who received continuous care in the NICU between January 2019 and March 2021 were recruited before discharge. Infants with congenital heart disease, severe respiratory disease, chromosomal abnormality, primary immunodeficiency disease, or an allergy to LAB or soybeans were excluded from this study. Written informed consent was obtained from the guardians of all infants involved in the study after explaining the study to them. All of the guardians reported their family histories of allergic diseases using questionnaires.

**Figure 1 nutrients-16-03635-f001:**
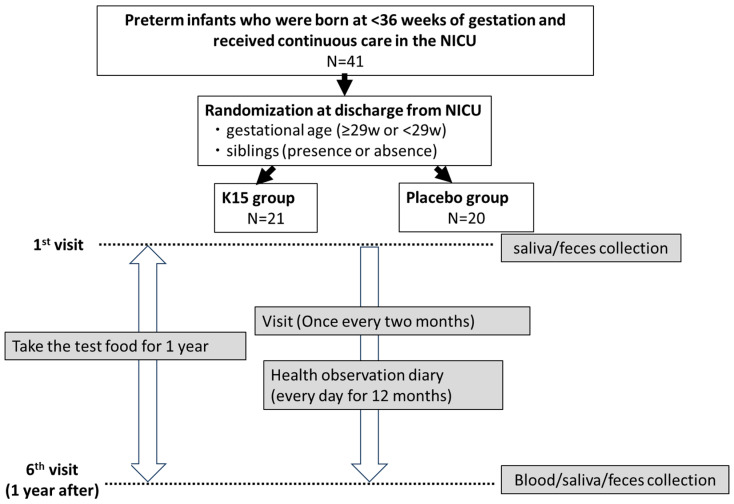
The flowchart describing this study.

Participants were randomly assigned to the K15 and placebo groups using a dynamic allocation method executed by an independent third party to ensure impartiality. The allocation was adjusted for two stratification factors: gestational age and the presence of siblings. Neonates were stratified into two categories based on their gestational age: those born at 29 weeks or more and those born at less than 29 weeks. Additionally, siblings’ presence or absence was considered as a second stratification factor, since infants are likely to contract respiratory infections transmitted from their siblings after NICU discharge. To maintain the double-blind nature of the study, neither the participants nor the healthcare providers were aware of the group assignments. This methodology was implemented to minimize potential biases and enhance the validity of the study results.

Infants received the test food once daily for one year, starting at the first visit after NICU discharge. All guardians recorded the following checklist items about the study infants each day: body temperature (≥37.5 °C or <37.5 °C, twice a day), consumption of test foods, foods including other LABs, medications (antipyretic analgesics, antibiotics, over-the-counter cold medicines, and other probiotics and postbiotics), symptoms, and hospital visits. The study infants received routine medical checks at regular visits every two months, and the physician checked their adverse events and hospitalizations for RTIs. The subjects’ clinical characteristics were collected at the beginning and end of the study.

### 2.3. Outcomes

The primary outcome was the number of febrile days (≥37.5 °C) during the test period. The secondary outcomes were as follows: (a) the incidence and severity of RTIs, as diagnosed by a physician; (b) the changes in serum immunoglobulin G (IgG), immunoglobulin A (IgA), and immunoglobulin M (IgM) levels between before and after the study period; and (c) the adverse events associated with K15 intake. The changes in s-IgA levels in saliva samples and bacterial microbiota in fecal samples before and after the study period were measured as exploratory outcomes.

### 2.4. Subgroup Analysis

The primary analysis of this study focused on comparing the number of febrile days between the K15 group and the placebo group. To further investigate factors influencing on effectiveness of K15 in preventing infections, we conducted subgroup analyses comparing the number of febrile days between the K15 group and the placebo group across the following subgroups: (1) High Adherence Rate Group: Participants with an adherence rate of 97% or higher, defined as the proportion of days they took the medication out of the days they were prescribed to take it; (2) Low Frequency of LAB Intake Group: Participants who consumed LAB supplements for less than 66 days (median consumption days across all participants) during the study period; (3) Subjects with Older Siblings Group: Participants who had older siblings; and (4) Nursery School Attendance at One Year Old Group: Participants who attended a nursery school at the age of one year.

### 2.5. Sample Collection

Before the commencement of the study, saliva and fecal samples were collected during the initial outpatient visit following discharge. Additional samples were collected at the end of the 12-month study period. Saliva samples were collected using Salivette^®^ tools (SARSTEDT AG & Co. KG, Nümbrecht, Germany) and centrifuged. Fecal samples were collected at home by the subjects’ guardians before the outpatient visit, brought to the hospital, and transferred to storage tubes. All samples were immediately frozen and stored at −80 °C.

### 2.6. Measurement of s-IgA Levels in Saliva Samples

s-IgA levels in saliva samples were determined using a YK280 Human s-IgA (Saliva) ELISA Kit (Yanaihara Institute Inc., Shizuoka, Japan). Sample preparation and ELISA were performed according to the manufacturer’s protocol.

### 2.7. Analysis of Gut Microbiota

Fecal samples were lyophilized using a VD-250R Freeze Dryer (TAITEC, Saitama, Japan) and subsequently crushed using a multi-bead shocker (Yasui Kikai, Osaka, Japan) at 1500 rpm for 2 min. Lysis Solution F (Nippon Gene CO., LTD., Tokyo, Japan) was added to the crushed samples, which were incubated at 65 °C for 10 min. Following centrifugation at 12,000× *g* for 2 min, the supernatant was separated, and DNA was extracted using a Lab Aid 824s DNA Extraction kit (ZEESAN, Xiamen, China) according to the manufacturer’s protocol. Bacterial 16S rRNA gene fragments (V3–V4) were amplified from purified DNA using a two-step tailed PCR method. The primers used for first-round and second-round PCR were as follows: first forward, 5′-ACACTCTTTCCCTACACGACGCTCTTCCGATCT-NNNNN-CCTACGGGNGGCWGCAG-3′; first reverse, 5′-GTGACTGGAGTTCAGACGTGTGCTCTTCCGATCT-NNNNN-GACTACHVGGGTATCTAATCC-3′; second forward, 5′-AATGATACGGCGACCACCGAGATCTACAC-Index2-ACACTCTTTCCCTACACGACGC-3′; and second reverse, 5′-CAAGCAGAAGACGGCATACGAGAT-Index1-GTGACTGGAGTTCAGACGTGTG-3′. PCR amplicons were sequenced using the MiSeq system and MiSeq Reagent Kit v3 (Illumina, San Diego, CA, USA) at Bioengineering Lab. Co., Ltd (Sagamihara, Japan) [15].

Regarding fecal microbial analyses, read sequences were extracted using the fastx_barcode_splitter tool of the FASTX-Toolkit (ver. 0.0.14) if the beginning of the read sequence exactly matched the primer sequence used. Primer sequences were removed from the extracted reads using fastx_trimer of the FASTX-Toolkit. Sequences with a quality value below 20 were removed using sickle (ver. 1.33), and sequences shorter than 130 bases and their paired sequences were discarded. The paired-end read joining script FLASH (ver. 1.2.11) was used to join the reads. The Dada2 plugin of Qiime2 (ver. 2022. 8) was employed to remove chimeric sequences and noise, and representative sequences and ASV tables were outputted [16]. Representative sequences were compared with the EzBioCloud 16S database using the feature classifier plugin to estimate phylogeny. Linear discriminant analysis (LDA) of effect size (LEfSe, ver. 1.0.8) was used for between-group comparison analysis to test for differences in relative abundance among the groups [16]. Statistical analysis was performed following Segata et al. [17]. The diversity plugin of Qiime2 was used to perform α-diversity and β-diversity analyses [16]. Functional prediction analysis involved the generation of functional composition tables based on METACYC (pathway) using picrust2 (ver. 2.3.0.b) with the frequency table output from Qiime2 and representative sequences as inputs. Principal coordinate analysis (PCoA) was performed using the diversity and emperor plugins of Qiime2, whereas statistical analysis of the METACYC (pathway) results was performed using STAMP (ver. 2.1.3) [18].

### 2.8. Statistical and Bioinformatic Analyses

This study utilized a double-blind, randomized controlled trial (RCT) design, focusing on two groups: the K15 group and the Placebo group. Statistical analysis was performed to compare these groups. For the primary endpoint, the number of febrile days, a *t*-test was employed to compare the groups. This analysis provided the mean values, 95% confidence intervals (CI), and *p*-values for the comparisons. Subgroup analysis of the number of febrile days was also conducted using *t*-tests to compare the groups. Different statistical tests were used for other analyses, depending on the data type. Continuous variables were compared between groups using either the *t*-test or Wilcoxon’s rank-sum test. Fisher’s exact test was used to examine the independence between the two groups for categorical variables. Summary statistics were reported: continuous variables were described by their means and 95% CIs or standard deviations, while categorical variables were presented as frequencies and percentages. All analyses were performed using SAS version 9.4 (SAS Institute Inc., Cary, NC, USA).

## 3. Results

### 3.1. Baseline Characteristics of Participants

Forty-one infants were enrolled in this study, with twenty-one and twenty randomly assigned to the K15 and placebo groups, respectively. Three infants did not attend follow-up visits, and the data associated with the remaining thirty-eight infants (K15: twenty infants, placebo: eighteen infants) were analyzed. The median gestational weeks of the groups were 32.5 weeks [24 weeks–35 weeks] and 32 weeks [25 weeks–35 weeks], and the median birth weights of the groups were 1554 g [677 g–2296 g] and 1822 g [558 g–2139 g], respectively. Although more than half of infants had a history of mechanical ventilation including non-invasive ventilation in the NICU, most of them had been weaned from ventilation within a few days. There were no sources of bias in the participants’ backgrounds (Table 1).

### 3.2. Primary Outcome (Number of Febrile Days During the Study Period)

The results for the primary outcome are presented in Table 2.

The mean numbers of febrile days (≥37.5 °C) (mean and range) in the K15 and placebo groups were 4.5 [1.5–7.4] and 6.6 [2.6–10.5], respectively, with no difference found between the groups. The results of all subgroup analyses are shown in Appendix A. No significant differences were observed in any of the comparisons, although the number of febrile days tends to be lower in the K15 subgroup stratified by presence of older siblings.

### 3.3. Secondary Outcomes

The results as to the secondary outcomes are presented in Table 3.

The incidence of RTIs and the total number of days with an RTI did not significantly differ between the K15 and placebo groups (Table 3a). One infant in each group experienced RSV infection, which is known to be severe in preterm infants. Two infants in the placebo group were admitted to a hospital and treated with mechanical ventilation, but the incidence of severe RTIs did not differ significantly between the groups. Regarding humoral immunity, the changes in serum IgG, IgA, and IgM levels before and after the study did not differ between the groups (Table 3b). To evaluate the effect of K15 on specific antibody acquisition, we measured the titers of antibodies against hepatitis B, for which immunization is routinely performed in infancy. However, the mean titers were not significantly different between the two groups (Appendix A). Concerning the safety of K15 intake, the number of adverse events was not different between the two groups (Table 3c), and the adverse events could not be demonstrated to be associated with the study food intake.

### 3.4. s-IgA Levels in Saliva Samples

Total salivary s-IgA levels in the K15 and placebo groups before and after the study period are presented in Table 4. Total s-IgA levels in saliva samples after the study period were not significantly different between the K15 and placebo groups (33.3 ± 16.4 µg/mL vs. 31.8 ± 13.5 µg/mL). The change rates of s-IgA levels were similar between the groups (1.46 ± 1.02 vs. 1.75 ± 1.44).

### 3.5. 16S rRNA Gene Sequence Analysis of Fecal Samples

α-Diversity was higher after the study period in both groups, as described by rarefaction curves of Chao1 in Appendix A. After the intervention, no significant difference in α-diversity was observed between the two groups. Similarly, regarding β-diversity, the PCoA plot of the fecal microbiota based on the weighted UniFrac distance indicated that the microbial profiles differed between the points before and after the study period in each group. However, the K15 treatment did not lead to microbial profile separation from the placebo. The observed results were supported by PERMANOVA (Appendix A).

Changes in the fecal microbiota composition for each group at the genus level are presented in Figure 2. In both groups, *Bifidobacterium* (green in the figure), *Escherichia* (purple in the figure), and Enterobacteriaceae (blue in the figure) were predominant before the intervention. However, substantial changes in the microbiota were detected after the study period, although *Bifidobacterium* remained predominant. There were no differences in the microbial composition between the K15 and placebo groups before and after the intervention. As a result of LEfSe after the intervention, the family Sutterellaceae and the genera *Faecalimonas* and *Pseudoflavonifractor* were enriched in the K15 group. In contrast, the family Peptostreptococcus was enriched in the placebo groups (Figure 3). Functional prediction analysis based on METACYC pathways illustrated that the metabolism of succinate to butanoate was more robust in the K15 group, whereas butyrate synthesis was not higher in this group.

## 4. Discussion

This is the first double-blind, randomized, placebo-controlled trial of the heat-killed probiotic *P. acidilactici* K15 in preterm infants after discharge from the NICU. Although heat-killed *P. acidilactici* K15 is safe, there were no clear data to show that it is effective in preventing fever and respiratory infections in preterm infants. Subgroup analysis suggested that having older siblings may affect the efficacy of K15.

Several systematic reviews and meta-analyses reported that probiotics can reduce the severity, duration, and incidence of RTIs [19,20,21]. Some systematic reviews and meta-analyses in children, including infants, also described the beneficial effects of probiotics [22,23,24]. However, in these meta-analyses, the quality of evidence was low, and these studies differed regarding the study period, the type and dosage of probiotics, and the age of the subjects. Regarding preterm infants, only two clinical studies were identified. Luoto et al. conducted a randomized, double-blind, placebo-controlled study of 94 preterm infants. They revealed that the incidence of RTIs was lower during the 12-month follow-up period in those receiving LABs or galactooligosaccharide between days 3 and 60 after birth [25]. The other clinical study, conducted by Aryayev et al., revealed that 62 late preterm infants who took a probiotic *Escherichia coli* strain during the newborn phase had lower mean numbers of RTIs after 28 days, 6 months, and 12 months, compared to those treated with the placebo [26].

In this study, the mean number of febrile days (≥37.5 °C) was not significantly lower in the K15 group than in the placebo group. Moreover, the incidence and total days of RTIs and other severity factors were not significantly lower in the K15 group. This result could be related to the adherence to K15 intake and other confounding factors. Stratified analysis indicates a trend of fewer febrile days in the subgroup with a high K15 adherence rate/low LAB intake/nursery school attendance at one year old. A statistically significant difference was observed in febrile days between the K15 group and the placebo group among children with older siblings. These data were partially compatible with our previous report on the effect of K15 in preschool children [14]. Our study featured a long observation period of one year, within which a variety of confounding factors could easily have exerted influence. A larger population size and stricter adjustment for confounding factors might have yielded a more significant trend.

LABs modulate immune systems by activating innate immune cells. LABs are recognized in the small intestine by DCs and macrophages, leading to the activation of overall innate immunity, including natural killer cells. Moreover, T and B cells’ functions are subsequently upregulated by various cytokines and costimulatory molecules provided by DCs and macrophages; thus, LABs also affect acquired immunity [27]. The IgA level is an important indicator of immune activation for innate and acquired immunity. In our previous in vitro study, K15 promoted IgA production by B cells through IL-6 and IL-10 secreted by myeloid DCs [13].

Moreover, in clinical studies of children and adults, K15 also promoted the secretion of salivary s-IgA, which has a significant role in preventing infection at mucosal sites [13,14]. In this study, the changes in serum IgG and salivary s-IgA levels did not significantly differ between the two groups during the study period. This result may be attributable to the small number of study subjects and the immune immaturity of preterm infants compared to children and adults.

Intestinal microbiota development in infants is influenced by various factors, including the mode of delivery (transvaginal delivery or cesarean section) [28], diet (human milk or formula, and type of weaning food) [29], infection and antibiotic use [30], and host genes [31]. *Bifidobacterium* is usually the predominant bacterium in the early period in infants, whereas *Clostridium* and *Bacteroides*, among others, become predominant after the weaning period [32,33]. LABs are more likely to colonize infants transvaginally delivered or infants who consume human milk. Several studies have illustrated the beneficial effect of LABs as a probiotic used to maintain a *Bifidobacterium*-enriched microbiota in infants [34,35]. A few previous studies have reported the effects of heat-killed LABs on the intestinal microbiota of infants and children. Canani et al. reported that the heat-killed strain *Lactobacillus paracasei* CBA L74 promoted the development of butyrate producers such as Ruminococcaceae, resulting in an increase in the relative abundance of genes predicted to be involved in butyrate synthesis [36]. Using LEfSe, the current study demonstrated that the genus *Faecalimonas* was enriched at the end of the study in the K15 group. Within this genus, *F. umbilicata* isolated from human feces produces acetic acid and vitamin B12, and it can promote butyrate production by butyrate-producing bacteria such as *Faecalibacterium prausnitzii* [37]. *Faecalibacterium* strains stimulate the immune system and T cell differentiation, and can prevent infection, as previously reported [38]. We could not observe an increase in butyrate synthesis via functional prediction analysis. The effect of K15 on the production of short-chain fatty acids, including butyric acid, in preterm infants needs to be further investigated in future studies.

The safety of oral postbiotics was demonstrated in previous studies [39], including reports using heat-killed K15 [13,14]. There is a risk of translocation of live probiotics given to premature babies admitted to the NICU (FDA warning: https://www.fda.gov/news-events/press-announcements/fda-raises-concerns-about-probiotic-products-sold-use-hospitalized-preterm-infants accessed on 24 October 2024); however, postbiotics could reasonably be expected to have a better safety profile than probiotics, as the microorganisms they contain have lost the capacity of translocation. It is meaningful that this study has contributed evidence of the safety of heat-killed K15 as a postbiotic in preterm infants.

This study had several limitations. First, although subjects were randomly divided into two groups, the sample size was small, which precluded stratified analyses with adjustment for clinical factors. This study was an exploratory study, and additional large-scale studies are needed. Also, each packet of K15 contained 5 × 10^10^ bacteria, which permitted it to be taken by infants once daily, so we could not evaluate a high dose of K15. Second, as saliva and fecal samples were only collected before and after the study, we did not assess the trajectory of salivary s-IgA level and fecal microbiota changes. Moreover, because of the small amount of feces obtained, we could not analyze gut metabolites, which is essential for determining the mechanism of effect of the probiotic. Finally, the COVID-19 pandemic in 2020 and associated control measures and behavioral alterations appear to have affected the incidence of several RTIs, so the effectiveness of K15 might not be accurately assessed. Further detailed studies are needed.

## 5. Conclusions

Our study revealed no significant differences in the number of febrile days and some indicators of episodes of RTIs between the K15 and placebo groups. In subgroup analysis, the number of febrile days tends to be lower in the K15 subgroup stratified by presence of older siblings. Meanwhile, the enrichment of *Faecalimonas* in the K15 group might increase butyrate production by butyrate-producing bacteria. Further large-scale and detailed studies are needed to determine the effectiveness of heat-inactivated K15 in preventing RTI in premature infants after discharge from the NICU.

## Figures and Tables

**Figure 2 nutrients-16-03635-f002:**
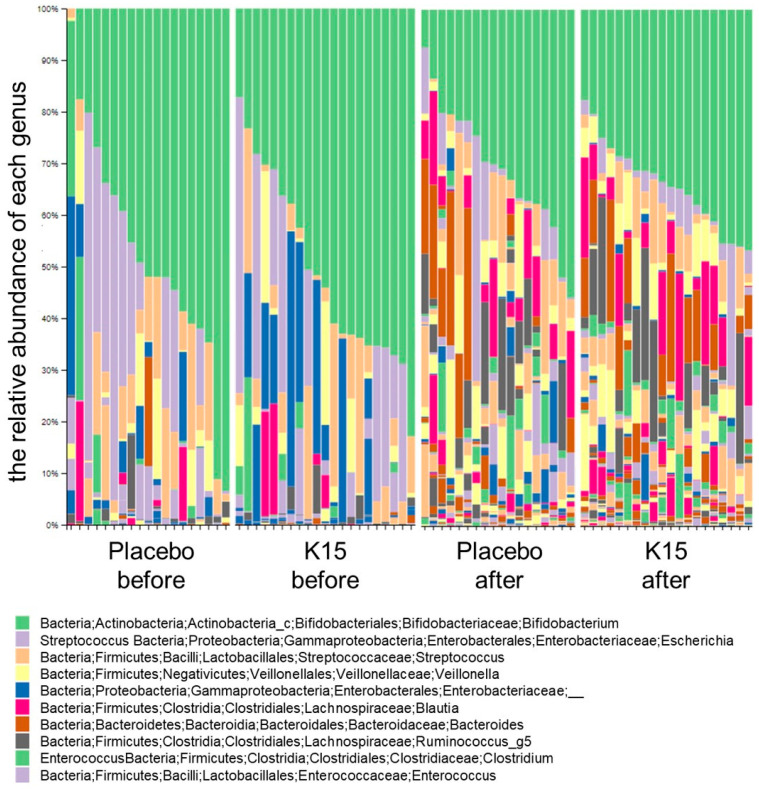
Changes in the fecal microbiota composition in each group at the genus level. The relative abundance ratios for each infant before and after treatment in the placebo group and K15 group. The vertical axis presents the relative abundance of each genus by color.

**Figure 3 nutrients-16-03635-f003:**
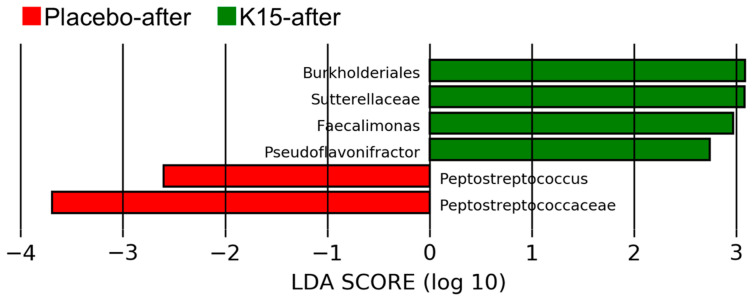
Linear discriminant analysis (LDA) scores of the fecal microbiota for each group after the study period. The horizontal axis presents the LDA scores for the placebo (red bars) and K15 groups (green bars) for bacterial groups with significant differences in abundance between points before and after the study period.

**Table 1 nutrients-16-03635-t001:** Baseline characteristics in the K15 and placebo groups.

		K15 (*n* = 20)	Placebo (*n* = 18)	*p*
Male, n (%)		12 (60.0)	12 (66.7)	0.7449
Female, n (%)		8 (40.0)	6 (33.3)	
Gestational weeks, median [range]		32.5 [24–35]	32 [25–35]	0.8972
Birth weight (g), median [range]		1554 [677–2296]	1822 [558–2139]	0.5006
Older siblings, n (%)		10 (50.0)	8 (44.4)	0.7568
Familial history of allergic diseases, n (%)		15 (75.0)	10 (55.6)	0.3071
History of RDS, n (%)		5 (25.0)	6 (33.3)	0.7240
History of mechanical ventilation including NIV, n (%)		12 (60.0)	11 (61.1)	>0.999
Days of mechanical ventilation, median [range]		2 [0–109]	3.5 [0–94]	0.8990
Nutrition before starting baby food	Exclusive breastfeeding, *n* (%)	5 (25.0)	4 (22.2)	0.9074
	Combination, n (%)	3 (15.0)	4 (22.2)	
	Exclusive formula, n (%)	12 (60.0)	10 (55.6)	
Days of discharge, median [range]		39 [11–166]	39.5 [11–132]	0.8397
Starting age of weaning in months, median [range]		7 [5–10]	7 [5–9]	0.9573
Nursery school attendance at one year old, *n* (%)		6 (30.0)	6 (33.3)	>0.999

RDS: respiratory distress syndrome; NIV: non-invasive ventilation.

**Table 2 nutrients-16-03635-t002:** The mean number of febrile days (≥37.5 °C) in the K15 and placebo groups.

Group	N	Mean	SD	Difference (K15-Placebo)			
				Mean	SD	95% CI	*p*-Value
K15	20	4.5	1.4	−2.1	7.1	[−6.8, 2.6]	0.2508
Placebo	18	6.6	1.9				

CI: confidence interval.

**Table 3 nutrients-16-03635-t003:** Secondary outcomes in this study.

**(a) Incidence and Severity of Respiratory Tract Infections in Each Group**									
					**K15 (*n* = 20)**		**Placebo (*n* = 18)**		** *p* **
Incidence of respiratory tract infections, *n* (%)					20 (100)		16 (88.9)		0.2176
Total days of respiratory tract infections, median [range]					41.5 [2–129]		35.5 [0–334]		0.8378
Respiratory syncytial virus infection, *n* (%)					1 (5.0)		1 (5.6)		>0.9999
Hospital visit, *n* (%)					19 (95.0)		16 (88.9)		0.5946
Total days of hospital visits, median [range]					4.5 [0–17]		4.5 [0–29]		0.9066
Hospital admission with respiratory tract infection, *n* (%)					0 (0.0)		2 (11.1)		0.2065
Oxygen use, n (%)					0 (0.0)		2 (11.1)		0.2065
Mechanical ventilation, *n* (%)					0 (0.0)		2 (11.1)		0.2065
**(b) The changes in serum** **IgG, IgA, and IgM levels between, before and after the study period in each group**									
	**IgG, mg/dL (mean ± SD)**			**IgA, mg/dL (mean ± SD)**			**IgM, mg/dL (mean ± SD)**		
	**Before**	**After**	** *p* **	**Before**	**After**	** *p* **	**Before**	**After**	** *p* **
K15 (*n* = 20)	383 ± 189	566 ± 129	0.4882	<10.0 ± 0.0	27.6 ± 13.4	0.2509	23.7 ± 12.4	81.5 ± 29.3	0.5094
Placebo (*n* = 18)	371 ± 209	602 ± 184		<10.0 ± 0.0	35.0 ± 24.6		21.4 ± 5.9	67.7 ± 26.7	
**(c) Adverse effects in each group during the study period**									
					**K15 (*n* = 20)**		**Placebo (*n* = 18)**		** *p* **
Gastrointestinal tract symptoms, *n* (%)					13 (65.0)		12 (66.7)		>0.9999
Total days with gastrointestinal tract symptoms, mean ± SD					12.4 ± 25.5		7.8 ± 17.9		0.8474
Skin symptoms, *n* (%)					11 (55.0)		6 (33.3)		0.2097
Total days with skin symptoms, mean ± SD					4.2 ± 7.8		3.1 ± 6.3		0.6000

IgG: immunoglobulin G; IgA: immunoglobulin A; IgM: immunoglobulin M. SD: standard deviation.

**Table 4 nutrients-16-03635-t004:** s-IgA levels in saliva samples before and after the study period in the K15 and placebo groups.

	K15 (*n* = 20)	Placebo (n = 18)	*p*
Before	34.6 ± 32.1	24.9 ± 16.2	0.2448
After	33.3 ± 16.4	31.8 ± 13.5	0.7713
Change rate (after/before)	1.46 ± 1.02	1.75 ± 1.44	0.4734

Data are presented as µg/mL (mean ± SD). s-IgA, secretory immunoglobulin A.

## Data Availability

Data supporting reported results can be requested from the principal author.

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
