# Peer review of "The Effectiveness of Heat-Killed Pediococcus acidilactici K15 in Preventing Respiratory Tract Infections in Preterm Infants: A Pilot Double-Blind, Randomized, Placebo-Controlled Study"

_nutrients, 2024, doi:10.3390/nu16213635_

Round 1
Reviewer 1 Report
Comments and Suggestions for Authors
The article under review, titled The Effectiveness of Heat-Killed Pediococcus acidilactici K15 in Preventing Respiratory Tract Infections in Preterm Infants: A Pilot Double-Blind, Randomized, Placebo-Controlled Study, conducted by Kenichi Takeshita et al., investigates the impact of a specific bacterial strain in preterm infants, aiming to prevent respiratory tract infections (RTIs).
The study adopts a rigorous double-blind, randomized, placebo-controlled design, which increases the reliability of the results. The randomization and blinding minimize biases that could compromise the validity of the findings.Despite the limited number of participants (41 infants), the study uses a relatively robust sample for an exploratory pilot study, especially considering the delicate nature of the target population (preterm infants). Although a subgroup analysis suggested that K15 might be more effective among children with older siblings (p=0.0482), this finding should be viewed with caution due to the small sample size and the multiplicity of tests. Results based on small subgroups are at risk of being influenced by biases or random factors.
This study, while innovative and safe, presents methodological and statistical power limitations. Although there is a favorable trend for the intervention with Pediococcus acidilactici K15, particularly in subgroups, larger and more controlled clinical trials are needed to validate its efficacy. Additionally, the impact of the intestinal microbiota and the mechanisms of action of K15, mentioned as potential influencers, deserve further investigation in future studies.
Comments on the Quality of English LanguageThe English is clear, precise, and well-structured, making the complex research easy to follow.
Author Response
RESPONSES TO REVIEWERS’ COMMENTS
We wish to express our heartfelt gratitude to you for their insightful comments, which have helped us to significantly improve our manuscript. Please find our responses to the reviewers' comments and suggestions listed below.
Changes in the revised manuscript are highlighted in yellow.
Comments and Suggestions for Authors
The article under review, titled The Effectiveness of Heat-Killed Pediococcus acidilactici K15 in Preventing Respiratory Tract Infections in Preterm Infants: A Pilot Double-Blind, Randomized, Placebo-Controlled Study, conducted by Kenichi Takeshita et al., investigates the impact of a specific bacterial strain in preterm infants, aiming to prevent respiratory tract infections (RTIs).
The study adopts a rigorous double-blind, randomized, placebo-controlled design, which increases the reliability of the results. The randomization and blinding minimize biases that could compromise the validity of the findings.Despite the limited number of participants (41 infants), the study uses a relatively robust sample for an exploratory pilot study, especially considering the delicate nature of the target population (preterm infants). Although a subgroup analysis suggested that K15 might be more effective among children with older siblings (p=0.0482), this finding should be viewed with caution due to the small sample size and the multiplicity of tests. Results based on small subgroups are at risk of being influenced by biases or random factors.
This study, while innovative and safe, presents methodological and statistical power limitations. Although there is a favorable trend for the intervention with Pediococcus acidilactici K15, particularly in subgroups, larger and more controlled clinical trials are needed to validate its efficacy. Additionally, the impact of the intestinal microbiota and the mechanisms of action of K15, mentioned as potential influencers, deserve further investigation in future studies.
Reply: We appreciate the reviewer's comment. We agree with the reviewer that small sample size has a risk of being influenced by biases or random factors especially in the subgroup analysis. We have included the data from the subgroup analysis in the supplementary materials (Supplementary Table 1) and have stated that the results of the subgroup analysis should be used for reference. We would like to argue that we would like to argue that a larger and properly planned clinical trial is necessary to verify the usefulness of K15, and we have described it in the conclusion section.

Reviewer 2 Report
Comments and Suggestions for Authors
- This is a rather generic topic studied regarding the potential efficacy and safety of heat-killed P. acidilactici K15 in relation to respiratory tract infections. However, it suffers from a very small sample size, insufficient statistical analysis, over-interpretation of results, and lack of controlling for confounding factors (which, the authors have only briefly acknowledged in the discussion section). It is applaudable for an attempt to show a negative finding, which in itself, is useful information - potentially, the paper could be rephrased and presented to highlight the lack of any significant finding instead of honing in on the few (coincidental) positive findings that have neither statistical nor clinical significance (or even a biologically plausible justification). Too much of the 'evidence' in this paper relies on the discussion's section citing of other papers, but not much (arguably none) of it is demonstrated in this paper.
- Sample size seems rather small. Has a power calculation been performed to ensure that it is sufficient in both exposure and control groups? Further to this, the stratification / subgroup sample sizes are even smaller, so there is need for a power calculation to justify the minimum sample size. These should be reported in the methods section.
- P.6 L.193: In subgroup analysis, the number of febrile days tends to be lower in the K15 subgroup with a high adherence rate than in the placebo group. There is no statistical evidence for this from table 2. Even the group of ‘subjects with other siblings’ has a confidence interval that is not significant, so how was a (marginally) significant p-value obtained? This calls into question the validity of calculation performed to obtain the result. Even if this were a statistically significant finding, what biological reason drove the choice of this categorisation? Were variables analysed at random fishing for a positive result? Such an exploratory approach should be outlined and justified in the methods section. Further to this, if such an approach was chosen, the fact that multiple tests were conducted in search for a positive finding is only justifiable if a multiple test correction (fda, Bonferroni, etc.) is conducted, which is absent here. Correcting for a multiple test penalty would surely make even the positive finding here clearly an insignificant finding.
- P.8 L.243: This set of statements are counterintuitive, as they contradict the primary statement of 'no difference in composition'. If there is a difference, as reported in the latter sentences and the LefSe LDA score figure, the accompanying statistical evidence should be reported (as before, the multiple test penalty application here is necessary to show significance after so many tests). "There was no difference in the microbial composition between the K15 and placebo groups before and after the intervention. As a result of LEfSe after the intervention, the family Sutterellaceae and the genera Faecalimonas and Pseudoflavonifractor were enriched in the K15 group. In contrast, the family Peptostreptococcus was enriched in the placebo groups (Figure 3)." Further to this, was there a functional / practical application of these differences, or are they merely a descriptive statistic (which is to be expected in such metagenomic analyses)?
- P.9 L.264: This is a false conclusion from the data: "... subgroup analysis suggests the potential effect and safety of P. acidilactici K15 in reducing RTIs in infants discharged from NICU." As commented above, there is NO statistical evidence for this, and the purported 'trend' is only visible in specifically picked subgroups with no biological rhyme or reason. There is no biologically plausible mechanism for supporting this relationship (with older sibling presence), and further, there is no reason to believe that this is evidence of efficacy. In terms of safety, at best, there is evidence of non-adverse effects (not worse) than placebo, but this is not the same as evidence of safety conferred e.g. reduction in adverse effects.
- A more appropriate test to show relationships between these tests is a multivariate test to quantify the relative effect sizes between tested variables. Conducting several univariate analyses as done here is open to various biases, false positives, and implicit confounders that are not accounted for. If this was attempted, both methods and results should make it clearer, as the only mention is that t-tests and Wilcoxon's rank-sum tests were used, but it is not apparent which results are reported in which sections.
- P.10 L. 313: This reference requires the year of publication. "Canani et al. reported that the heat-killed probiotic strain Lactobacillus paracasei CBA L74 pro-314 moted the development of butyrate producers such as Ruminococcaceae..."
- The conclusion section over-interprets the statements "... the number 345 of febrile days was fewer among K15-treated infants..." as there is no statistical evidence for this (and the supposed 'trend' is not apparent when considering the small sample size and effect size involved). Similarly, there is also no evidence for safety or further research (if anything, the negative findings suggest no further research should be pursued) "We concluded that K15 is safe and has the potential for further large-scale research for RTI prevention in pre-term infants after discharge from NICU."
Author Response
RESPONSES TO REVIEWERS’ COMMENTS
We thank the reviewer for the careful reading of our manuscript and critical comments and suggestions. Guided by these suggestions, we have prepared a revised version of the manuscript. The revised parts in the manuscript are highlighted in yellow. A point-by-point response to all comments made by the reviewers is found below, where reviewers’ comments are italicized in boldface.
- This is a rather generic topic studied regarding the potential efficacy and safety of heat-killed P. acidilactici K15 in relation to respiratory tract infections. However, it suffers from a very small sample size, insufficient statistical analysis, over-interpretation of results, and lack of controlling for confounding factors (which, the authors have only briefly acknowledged in the discussion section). It is applaudable for an attempt to show a negative finding, which in itself, is useful information - potentially, the paper could be rephrased and presented to highlight the lack of any significant finding instead of honing in on the few (coincidental) positive findings that have neither statistical nor clinical significance (or even a biologically plausible justification). Too much of the 'evidence' in this paper relies on the discussion's section citing of other papers, but not much (arguably none) of it is demonstrated in this paper.
Reply: Thank you for reviewing our manuscript. This study was planned following the effects and safety of K15 in preschool children (Nutrients 2020;12:1). As it was not possible to predict the effects of K15 on small infants discharged from the NICU, we conducted the study as a pilot double-blind randomized placebo-controlled trial. As explained in the discussion section, the number of participants was relatively small partly due to the impact of the pandemic of the novel coronavirus infection (COVID-19). COVID-19 pandemic also may have had influence on febrile episodes in the study. Although the evidence that K15 is not effective in preventing respiratory infection for infants after leaving the NICU, we believe that it is worth conducting a larger-scale study based on the interpretation that K15 may reduce the incidence of respiratory infections in children under certain conditions. However, we agree with the reviewers' criticisms. We have only shown the main outcome in Table 2, and the subgroup analysis is shown in Table S2 as a supplementary material. Furthermore, we stated that the results of the subgroup analysis were negative (lines 220-221).
- Sample size seems rather small. Has a power calculation been performed to ensure that it is sufficient in both exposure and control groups? Further to this, the stratification / subgroup sample sizes are even smaller, so there is need for a power calculation to justify the minimum sample size. These should be reported in the methods section.
Reply: Thank you for giving us the opportunity to clarify these points. Because we were unable to predict the effectiveness of K15, we conducted a pilot study to obtain information on the effectiveness of K15 in preventing respiratory infections in children who had been discharged from the NICU. Based on the results of this study, the power was 15%. If we were to demonstrate the effect of K15 with the same frequency of fever as in this study, a total of 350 subjects would be required (power 80%).
- P.6 L.193: In subgroup analysis, the number of febrile days tends to be lower in the K15 subgroup with a high adherence rate than in the placebo group. There is no statistical evidence for this from table 2. Even the group of ‘subjects with other siblings’ has a confidence interval that is not significant, so how was a (marginally) significant p-value obtained? This calls into question the validity of calculation performed to obtain the result. Even if this were a statistically significant finding, what biological reason drove the choice of this categorisation? Were variables analysed at random fishing for a positive result? Such an exploratory approach should be outlined and justified in the methods section. Further to this, if such an approach was chosen, the fact that multiple tests were conducted in search for a positive finding is only justifiable if a multiple test correction (fda, Bonferroni, etc.) is conducted, which is absent here. Correcting for a multiple test penalty would surely make even the positive finding here clearly an insignificant finding.
Reply: Based on past research papers, we conducted a subgroup analysis of the factors that affect the frequency of infection. The factors we examined included the rate of consumption of the test drug, the frequency of consumption of other lactic acid bacteria preparations, whether or not the child had siblings to assess the risk of infection, and whether or not the child lived in a group setting. Although the analysis of these factors did not achieve sufficient statistical significance, it was thought that there was a possibility that the effect of K15 would become clearer in infants with a high risk of infection, such as those with many siblings or those living in a group setting. As multiple testing was not performed, there is no evidence of a statistically significant difference.
- P.8 L.243: This set of statements are counterintuitive, as they contradict the primary statement of 'no difference in composition'. If there is a difference, as reported in the latter sentences and the LefSe LDA score figure, the accompanying statistical evidence should be reported (as before, the multiple test penalty application here is necessary to show significance after so many tests). "There was no difference in the microbial composition between the K15 and placebo groups before and after the intervention. As a result of LEfSe after the intervention, the family Sutterellaceae and the genera Faecalimonas and Pseudoflavonifractor were enriched in the K15 group. In contrast, the family Peptostreptococcus was enriched in the placebo groups (Figure 3)." Further to this, was there a functional / practical application of these differences, or are they merely a descriptive statistic (which is to be expected in such metagenomic analyses)?
Reply: The relative abundance of each genus in the placebo and K15 groups was compared before and after the intervention. The bacterial composition of both groups changed before and after the intervention, but no differences were observed between the placebo and K15 groups. Lefse analysis showed that there were differences between the two groups in some bacteria after the intervention. Pathway analysis suggested that the K15 group may have increased production of short-chain fatty acids. However, this analysis did not directly measure the metabolites produced by the bacteria. The effects of K15 intake on the intestinal flora and metabolites are considered to be a topic for future research, and are described in the discussion section (lines 334-336).
- P.9 L.264: This is a false conclusion from the data: "... subgroup analysis suggests the potential effect and safety of P. acidilactici K15 in reducing RTIs in infants discharged from NICU." As commented above, there is NO statistical evidence for this, and the purported 'trend' is only visible in specifically picked subgroups with no biological rhyme or reason. There is no biologically plausible mechanism for supporting this relationship (with older sibling presence), and further, there is no reason to believe that this is evidence of efficacy. In terms of safety, at best, there is evidence of non-adverse effects (not worse) than placebo, but this is not the same as evidence of safety conferred e.g. reduction in adverse effects.
Reply: We agree with the reviewer’s comment. We changed the summary sentence to “This is the first double-blind, randomized, placebo-controlled trial of the heat-killed probiotic P. acidilactici K15 in preterm infants after discharge from the NICU. Although heat-killed P. acidilactici K15 is safe, there was no clear data to show that it is effective in preventing fever and respiratory infections in preterm infants. Subgroup analysis suggested that having older siblings may affect the efficacy of K15..” (lines 272-276). We also changed abstract according to the reviewer’s comments.
- A more appropriate test to show relationships between these tests is a multivariate test to quantify the relative effect sizes between tested variables. Conducting several univariate analyses as done here is open to various biases, false positives, and implicit confounders that are not accounted for. If this was attempted, both methods and results should make it clearer, as the only mention is that t-tests and Wilcoxon's rank-sum tests were used, but it is not apparent which results are reported in which sections.
Reply: We appreciate the reviewer's comments. We did not perform multivariate analysis because the number of subjects was relatively small. We have deleted the description of the subgroup analysis, which was based on insufficient evidence, and only discussed the interpretation in the discussion section.
- P.10 L. 313: This reference requires the year of publication. "Canani et al. reported that the heat-killed probiotic strain Lactobacillus paracasei CBA L74 pro-314 moted the development of butyrate producers such as Ruminococcaceae..."
Reply: The reference is indicated as reference #35 in the original MS.
- The conclusion section over-interprets the statements "... the number 345 of febrile days was fewer among K15-treated infants..." as there is no statistical evidence for this (and the supposed 'trend' is not apparent when considering the small sample size and effect size involved). Similarly, there is also no evidence for safety or further research (if anything, the negative findings suggest no further research should be pursued) "We concluded that K15 is safe and has the potential for further large-scale research for RTI prevention in pre-term infants after discharge from NICU."
Reply: We appreciate the reviewer's comments. Although no statistically significant difference was observed, even with a small number of subjects, the risk of infection was substantially reduced by having siblings, so we believe that the anti-infective effect of K15 can be expected. We changed the conclusion in accordance with the reviewer's comments, but we would like to argue that a properly planned clinical trial is necessary to verify the usefulness of K15 in preterm infants discharged from the NICU. (lines 358-364)

Reviewer 3 Report
Comments and Suggestions for Authors
Thank you for submitting the manuscript "The Effectiveness of Heat-Killed Pediococcus acidilactici K15 in Preventing Respiratory Tract Infections in Preterm Infants: A Pilot Double-Blind, Randomized, Placebo-Controlled Study" to Nutrients. The researchers evaluated an inactivated strain of a potentially postbiotic microorganism in prematurely born babies. The work is interesting, but in my opinion there is confusion between the concepts of probiotics, which involve live microorganisms (Hill, 2014), and postbiotics, which are inactivated strains. And it seems to me that from a safety point of view, this work can only be carried out with this population, since it involves an inactivated strain, since this is a high-risk population. I believe that this needs to be corrected throughout the text so that the manuscript can be considered for publication.
- Line#56: Add more information: in what quantity was it administered and for how long? Is this the only study conducted with this strain in humans? I think a paragraph with an overview of these studies is extremely important.
- It is important for the authors to indicate why they hypothesized that administering a microorganism to preterm infants could bring some benefit. The introduction as a whole seems superficial from this point of view.
- Introduction: the manuscript manipulated a potentially postbiotic microorganism but at no time did it provide the official IASPP definition.
- M&M: Did these children receive vitamin A and/or iron supplementation? In many countries, this supplementation is standard procedure, even without prior dosage after 6 months of life.
- Line#71: I think it would be valid to include more characteristics of the population, such as are the babies low weight? Were they born with some critical health condition?
- Line#84: why brothers were considered as a variable. An explanation should be added.
- Figures and tables: all titles of figures and tables should be reformulated because they need to be independent of the body of the text and therefore should provide more information.
- Figure 1: postbiotics are not medicine.
- Line#308: I believe that there should be a "among others" here, since there are many sources of variation.
- Line#329: I believe that this translocation problem does not exist in the case of postbiotics and this should be discussed at this point, providing the most consistent justifications regarding the use of this work.
- Throughout the text, the authors use the term probiotic, but I believe that the correct term is postbiotic, since the microorganism was inactivated.
Comments on the Quality of English LanguageEnglish needs refinement.
Author Response
RESPONSES TO REVIEWERS’ COMMENTS
We wish to express our heartfelt gratitude to you for their insightful comments, which have helped us to significantly improve our manuscript. Please find our point-by-point responses to the reviewers' comments and suggestions listed below. Changes in the revised manuscript are highlighted in yellow.
Comments and Suggestions for Authors
The work is interesting, but in my opinion there is confusion between the concepts of probiotics, which involve live microorganisms (Hill, 2014), and postbiotics, which are inactivated strains. And it seems to me that from a safety point of view, this work can only be carried out with this population, since it involves an inactivated strain, since this is a high-risk population. I believe that this needs to be corrected throughout the text so that the manuscript can be considered for publication.
Reply: We appreciate the reviewer’s insightful comment on this matter. Upon review, we confirm that K15 used in this study is an inactivated strain. We agree that the term “postbiotics” is more appropriate, and have carefully reviewed the terminology, revising it accordingly throughout the manuscript. We have also added the term “heat-killed” before “K15” throughout the manuscript.
.
Line#56: Add more information: in what quantity was it administered and for how long? Is this the only study conducted with this strain in humans? I think a paragraph with an overview of these studies is extremely important
Reply: Thank you for pointing it out. The clinical studies conducted with heat-killed K15 in humans consist of only two, both of which are our studies (reference No. 13 and 14). We have added detailed information of the referenced study regarding the administration of K15. Specifically, heat-killed K15 was administered in a quantity of 5 × 1010 bacteria for a period of sixteen weeks. Moreover, we have also added some details of the reference studies in the previous paragraph.
It is important for the authors to indicate why they hypothesized that administering a microorganism to preterm infants could bring some benefit. The introduction as a whole seems superficial from this point of view.
Reply: Thank you for the comment. We have added the hypotheses and revised the sentences in the end of the Introduction section to clarify our research background (lines 61-66).
Introduction: the manuscript manipulated a potentially postbiotic microorganism but at no time did it provide the official IASPP definition.
Reply: Thank you for pointing it out. We have modified the Introduction section, added the description about IASPP definition, and also added the reference paper (lines 43-48).
M&M: Did these children receive vitamin A and/or iron supplementation? In many countries, this supplementation is standard procedure, even without prior dosage after 6 months of life.
Reply: Thank you for asking. Some medications (antipyretic analgesics, antibiotics, over-the-counter cold medicines, and other probiotics and postbiotics) were recorded in the CRF; however, all medications and supplement intakes were not recorded in this study. In Japan, these supplements are not routinely administered to all preterm infants.
Line#71: I think it would be valid to include more characteristics of the population, such as are the babies low weight? Were they born with some critical health condition?
Reply: Thank you for the suggestion. Although Table 1 shows baseline characteristics of the population, we had added the explanations in the Results section (lines 193-197).
Line#84: why brothers were considered as a variable. An explanation should be added.
Reply: Thank you for the comment. Siblings were considered as a variable because the presence of siblings could potentially influence the primary outcome (number of febrile days) and secondary outcome (the incidence and severity of RTIs). Infants are more likely to contract respiratory infections transmitted from their siblings after NICU discharge. We have added an explanation in the manuscript to clarify this point (lines 92-94).
Figures and tables: all titles of figures and tables should be reformulated because they need to be independent of the body of the text and therefore should provide more information.
Reply: Thank you for pointing it out. We have checked all the titles of figures and tables and revised them if needed.
Figure 1: postbiotics are not medicine.
Reply: We have changed the word, “medicine” to “the test food”.
Line#308: I believe that there should be a "among others" here, since there are many sources of variation.
Reply: We have added the phrase, “among others” to the sentence in accordance with the reviewer's comment,
Line#329: I believe that this translocation problem does not exist in the case of postbiotics and this should be discussed at this point, providing the most consistent justifications regarding the use of this work.
Reply: Thank you for point it out. We agree with you. We have added an explanation at this point in this paragraph (lines 341-343).
Throughout the text, the authors use the term probiotic, but I believe that the correct term is postbiotic, since the microorganism was inactivated.
Reply: We agree that the term “postbiotics” is correct, and have carefully reviewed the terminology, revising it accordingly throughout the manuscript. We have also added the term “heat-killed” before “K15” throughout the manuscript.

Round 2
Reviewer 3 Report
Comments and Suggestions for Authors
This reviewer would like to thank the authors for their efforts in making the suggested corrections. It was possible to notice that the quality of the manuscript has improved significantly and is now much clearer and better discussed. One small correction I would suggest is to add the issue of children's supplementation to the text. However, I believe it can be accepted for publication.